# Intersectional tension: a qualitative study of the effects of the COVID-19 response on survivors of violence against women in urban India

Mita Huq,[1] Tanushree Das,[2] Delan Devakumar,[1] Nayreen Daruwalla,[2] David Osrin  [1]

[1]Institute for Global Health, UCL, London, UK
[2]Program on Prevention of Violence Against Women and Children, Society for Nutrition, Education & Health Action, Mumbai, Maharashtra, India

**Correspondence to**
Dr David Osrin;
d.osrin@ucl.ac.uk

## ABSTRACT

**Objectives** There is a concern worldwide that efforts to address the SARS-CoV-2 pandemic have affected the frequency and intensity of domestic violence against women. Residents of urban informal settlements faced particularly stringent conditions during the response in India. Counsellors spoke with registered survivors of domestic violence in Mumbai, with two objectives: to understand how the pandemic and subsequent lockdown had changed their needs and experiences, and to recommend programmatic responses.

**Design** Qualitative interviews and framework analysis.

**Setting** A non-government support programme for survivors of violence against women, providing services mainly for residents of informal settlements.

**Participants** During follow-up telephone counselling with survivors of violence against women who had previously registered for support and consented to the use of information in research, counsellors took verbal consent for additional questions about the effects of COVID-19 on their daily life, their ability to speak with someone, and their counselling preferences. Responses were recorded as written notes.

**Results** The major concerns of 586 clients interviewed between April and July 2020 were meeting basic needs (financial stress, interrupted livelihoods and food insecurity), confinement in small homes (family tensions and isolation with abusers) and limited mobility (power imbalances in the home and lack of opportunity for disclosure and stress relief). A major source of stress was the increased burden of unpaid domestic care, which fell largely on women.

**Conclusion** The COVID-19 pandemic has increased the burden of poverty and gendered unpaid care. Finance and food security are critical considerations for future response, which should consider inequality, financial support, prioritising continued availability of services for survivors of violence and expanding access to social networks. Decision-makers must be aware of the gendered, intersectional effects of interventions and must include residents of informal settlements who are survivors of domestic violence in the planning and implementation of public health strategies.

### Strengths and limitations of this study

► Interviews were conducted with almost 600 survivors of violence.
► Existing relationships between survivors and counsellors.
► All participants were residents of urban informal settlements in which stringent measures to limit the spread of COVID-19 were implemented.
► The study did not examine incidence of domestic violence against women who were not already registered.
► Telephone interviews were limited by women's willingness to provide information.

organisations working to prevent violence against women became concerned that the measures might themselves give rise to or escalate domestic violence.[1–5] This concern was informed by three strands of evidence. The first was that states and civil society organisations noted large increases in contacts through domestic violence helplines and websites at the same time as the numbers of face-to-face consultations fell.[6–13] The second was that the conditions imposed by the pandemic, such as lockdown and income loss, were likely to increase household and interpersonal conflict.[14 15] The third was that public health emergencies have been shown repeatedly to be followed by increases in violence against women and girls.[16 17] Amid reports that the frequency and severity of domestic violence were increasing,[17–20] WHO and United Nations Women (UN Women) raised concerns that measures such as quarantine and lockdown could increase the risk of violence against women.[8 21]

These concerns were observable in India, where an increase in violence against women was recognised in the media and some studies.[22 23] The reports describe a relationship

## BACKGROUND

As governments implemented measures to address the spread of SARS-CoV-2,

between loss of economic prospects, separation from friends and family, and amplified gendered expectations as a result of lockdown measures. Lockdown measures were implemented suddenly and stringently, particularly in urban informal settlements (slums). Residents of such settlements, in which ~40% of Mumbai's households are located, were seen as hotspots for contagion because of density and overcrowding, insubstantial housing and insufficient water and sanitation.[24 25] An average of five people share one or two rooms, most rely on shared toilets, and around two-thirds need to leave the house to get water. The government of Maharashtra announced a lockdown in Mumbai on 23 March 2020, for an initial 21 days with subsequent extensions. Major informal settlements were quarantined as red zones,[26] their access barricaded, public toilets disinfected and a police-enforced curfew instituted. Residents were confined to their homes unless they faced medical emergencies, transport services and vehicular movement curtailed, and industry and retail outlets closed. Residents were screened for SARS-CoV-2 through door-to-door visits by public and private sector healthcare workers and centres providing free food and healthcare were set up in schools and halls.[25]

These measures were commended for preventing a surge in cases in areas where the potential spread of SARS-CoV-2 was particularly concerning,[25 27] but fear of institutional quarantine was common and accompanied by a degree of community stigma. The risk of domestic violence does not appear to have been considered. At the same time as survivors of violence were potentially confined with their abusers, organisations working to prevent and respond to violence suffered resource limitations,[28 29] reduced capacities and interrupted monitoring and evaluation activities.[8 21 30] The civil society work of the Society for Nutrition, Education and Health Action (SNEHA) illustrates these pressures. Since 2000, the SNEHA Programme on Prevention of Violence Against Women and Children has provided support to over 13 000 survivors of violence against women. Services are open to all—particularly residents of Mumbai's informal settlements—and are delivered from five community and three public hospital counselling centres. Support is available through a combination of in-house services and liaison. In-house human resources include community outreach workers, counsellors, legal advisors and lawyers, and psychologists. These are complemented by liaison with hospital practitioners, the police, District Legal Aid services, shelters and psychiatrists.

As the response to the pandemic unfolded, the programme moved counselling and crisis intervention services online. Existing helplines were augmented and survivors of violence could call or email counselling centres for assistance. A cadre of 300 volunteers gave out contact details in communities. At the same time, counsellors contacted their existing clients and offered to provide follow-up telephonically. SNEHA developed a protocol for this follow-up, based on existing safety protocols and emphasising confidentiality and the same client-centred approach followed in face-to-face sessions. It became clear that these telephone conversations could provide a rich understanding of women's experiences of domestic violence in urban informal settlements during lockdown. In discussion with counsellors, we designed a topic guide for qualitative interviews that they could integrate into their discussion with existing clients. Our objectives were to use the interviews to understand how the COVID-19 pandemic and subsequent lockdown changed women's needs and experiences, and to recommend programmatic responses to domestic violence in light of pandemics and the strategies used to mitigate them.

## METHODS

### Setting and participants

Our study explored the experiences of survivors of domestic violence in a specific context:[31] residents of urban informal settlements in Mumbai whose suffering as a consequence of domestic violence by either an intimate partner or another family member had led them to register for support. Only about 8% of women in these circumstances get as far as registering with a support organisation,[32] and the sample, therefore, represents a selected group. Interviewees were women clients who had been interacting with SNEHA counsellors prior to the lockdown and had developed relationships with them. They were all survivors of violence, having connected with the organisation after surviving domestic violence—physical, sexual, emotional or economic violence, coercive control or neglect—by intimate partners or family members. Since they were all survivors, our interest was in changes in their experience rather than in the incidence of violence against women who had not previously made contact.

### Data collection

Eleven female counsellors with postgraduate education in social work or psychology and between 2 and 10 years of experience of supporting clients from our centres conducted interviews after receiving project-specific training. Clients had registered initially through self-report, referral from agencies such as the police, clinicians or non-government organisations, or referral by SNEHA community organisers or volunteers trained to identify and engage with survivors. Participants were contacted by their counsellors for telephone follow-up between March and May 2020. Counsellors explained that, as part of their conversations, they were doing a study to understand women's experiences during the lockdown. With agreement from clients, they administered a semistructured interview about how COVID-19 had affected their daily life, family life, ability to speak with someone and their counselling preferences. Interviews were conducted during working hours. If a client did not answer the phone, nearby field staff, volunteers or women's group members would visit her and encourage her to contact her counsellor to indicate her availability. Counsellors

provided all clients with information on COVID-19 (transmission, social distancing, handwashing, masks, emotional hygiene, anxiety management, freedom to contact the organisation for advice). If a family member answered the phone (or it seemed likely to the counsellor that someone was listening in), they said that the call was part of the COVID-19 prevention programme and that they were interested in the family's emotional health. Counsellors recorded responses in field notes categorised according to the topic guide (online supplemental file). Interviews were conducted in Hindi and Marathi and subsequently summarised in English by bilingual team leaders. Interviews were part of counselling calls that lasted up to an hour. All potential identifiers were removed from the dataset.

## Data analysis

The familiarisation process began by reading all the records. Two authors approached the data independently (MH and DO). We used the framework approach to thematic analysis.[33] Increasingly used in health research,[34] this approach lends itself to studies by multidisciplinary teams with fairly specific objectives and questions, limited time for data collection and a predefined large sample.[35] Data are charted in a matrix in which rows represent interviewees and columns represent codes, and information can be compared across cases and codes. We identified some of these codes before the analysis and augmented them with codes that emerged.[33 36–39] The two authors who analysed the data compared their codes iteratively. The emergent themes were discussed over four iterations, with inputs from all authors. An indexing and data exploration process expanded this framework by inductively adding themes and subthemes.

## Patient and public involvement

Clients of our counselling centres for survivors of violence against women were first involved during telephonic discussions with counsellors. The research questions were a direct response to concerns about their safety and emotional health and well-being during the response to the COVID-19 pandemic, and sought to understand their experiences and preferences for services. Dissemination will involve personal feedback of the findings to clients during subsequent counselling sessions and feedback during regular community mobilisation meetings in informal settlements across the city.

## RESULTS
### Participants

Between 1 April 2020 and 30 July 2020, counsellors spoke with 720 clients. Our analysis draws on interviews with 586 women for whom complete background information was available. At the time of interview, 54% of respondents were in the first month of lockdown, 40% in the second and 6% in the third month or later. Of the existing clients who provided information, 46% had been seeing

**Table 1** Participant characteristics at registration for counselling

|  | n (%) |
| --- | --- |
| **Age (years)** | |
| Under 20 | 26 (4) |
| 20–29 | 262 (45) |
| 30–39 | 205 (35) |
| 40–49 | 56 (10) |
| 50+ | 37 (6) |
| **Education** | |
| None | 63 (11) |
| Primary | 78 (13) |
| Lower secondary | 97 (16) |
| Higher secondary | 226 (39) |
| Higher | 110 (19) |
| Unknown | 12 (2) |
| **Marital status** | |
| Married | 431 (73) |
| De facto | 16 (3) |
| Separated or divorced | 64 (11) |
| Widowed | 29 (5) |
| Unmarried | 45 (8) |
| **Children** | |
| None | 217 (37) |
| 1 | 149 (25) |
| 2 or more | 220 (38) |
| **Family arrangement** | |
| Nuclear | 342 (59) |
| Joint | 194 (33) |
| Extended | 23 (4) |
| Living alone | 19 (3) |
| Other | 8 (1) |
| **Religion** | |
| Hindu | 278 (47) |
| Muslim | 241 (41) |
| Other | 67 (12) |
| **Remunerated employment** | |
| None | 392 (67) |
| Student | 7 (1) |
| Informal sector | 127 (22) |
| Formal sector | 60 (10) |
| All | 586 (100) |

their counsellor for less than 6 months, 33% for up to a year, 12% for up to 18 months and 8% for longer periods. Table 1 shows that most women were in their 20s or 30s (80%) and had attended school to at least secondary level (74%). Most had been married at some point (92%) and

around two-thirds had children (63%). The largest group had been living in nuclear families at the time of initial consultation (59%) and the next largest group in joint families with their in-laws (33%). Most lived in informal settlements (zopadpatti) or former workers' accommodation (chawls) (73%). The major religions were about equally represented. Around one-third of women (32%) worked outside the home, usually in the informal sector.

The background to initial consultation for most women had been intimate partner violence (IPV) (70%). However, 44% had suffered violence by another family member and in 28% of cases violence came from both intimate partners and other family members. Family violence was predominantly emotional (87%), economic (63%) and physical (54%) and coercive control (46%). The primary concerns that women said had led them to consult were economic abuse, particularly feeling that they had been denied money (56%), physical violence (49%), being compelled to leave their home (32%), being threatened by their partner (25%), having had their property or possessions taken by family members (19%) and suffering sexual violence (17%). Although instances of violence were not always the triggers for consultation, 86% of women described experiences of emotional violence, 76% economic violence, 64% physical violence, 56% neglect, 56% controlling behaviours and 35% sexual violence. Most had survived more than one of these forms of violence (86%) and 22% all six forms.

## A matrix of stressors

What emerged from women's accounts of life during lockdown was a matrix of stressors that interacted with domestic violence: worry about basic needs, confinement in small homes, limited mobility, concern for children and alcohol. Lockdown had confined them to four general living arrangements: a nuclear family, a joint family with husband and in-laws, with members of a natal family or either alone or with children. At the time of interview, around half of respondents (52%) were living with their partner in either nuclear or joint families. Most women who were not had relocated to live with their natal family. This usually represented a longer-term strategy of separation, but for some a short visit had become longer as a result of lockdown. For women who were confined in a nuclear family with a husband who had been abusive at the time of initial consultation, 27% were still surviving violence. For women who were living with a husband and in-laws, 28% were still surviving intimate partner or family violence.

## Worry about basic needs

Dominant in women's narratives were problems meeting basic needs. Half of respondents expressed overwhelming concerns about money and food, and most women who said that they were managing were drawing on their own or their family's savings. Families living in informal settlements depend largely on a mosaic of livelihoods in the informal sector and most women said that the lockdown

had caused work to be suspended. Restrictions on movement meant that daily earners could not go to work: 'I used to do domestic work but am now unable to go to work. My employers have been calling me to say that because … Covid-19 cases are high here, they don't want me to join work for another six months. I'm worried about finances' (20–29 years, separated with one child and living with natal family; physical, emotional and economic IPV at first consultation).

A few days of lockdown led many families to draw on their limited savings. 'We're running out of savings. My husband's boss is not able to pay him right now but is providing him with the basics so he's staying at the job. But my kids and I are struggling to make ends meet' (30–39 years, married with four children and living with husband in nuclear family; physical, emotional, economic IPV). The only safety net was the government ration system, itself compromised by halts in supply and quarantine of communities. Families who were eligible for rations became more dependent on them, but the quantity and quality of rations and food in the shops was compromised by supply problems. 'The food resources have been insufficient, as well as the whole family is present and two meals for all become a little tough with this current crisis' (20–29 years, unmarried and living in natal family; physical IPV). Food insecurity, therefore, resulted from both lack of money and unavailability of food: 'All shops are closed. The area is sealed. We are managing with the rations we get from the ration shop. Other shops are closed. Around 15 days ago, a woman had come to ask for our contact numbers to provide help, but we haven't heard from her or anyone so far' (20–29 years, unmarried and living in natal family; physical family violence). Coupled with inconsistent or insufficient government and non-government assistance, these pressures made members of food insecure families leave the home in contravention of lockdown policy to seek food, exposing themselves to SARS-CoV-2 and punitive action by law enforcement. The combination of insecurity and violence was a potent stressor: 'Barely being able to manage. My husband came and fought with me a lot a couple of days ago. After that now the shop below my house is giving me stuff on loan' (20–29 years, married with three children and living with children; physical, sexual, emotional, economic IPV).

## Confinement in small homes

Stress was increased by the small living spaces in informal settlements in which most respondents lived. Lack of privacy and time alone exacerbated tension in large families, whereas loneliness was reported by women who lived alone or were being forcibly isolated or neglected. Tension was often increased by a combination of overcrowding and precarity. '… I'm in tension and stress due to lack of money and resources. Sometimes we fight with each other, but … I'm not able to go outside…' (40–49 years, married with six children and living with husband in nuclear family; physical, sexual, emotional, economic

IPV). Women living in joint families faced predominantly emotional violence and physical violence in three cases.

Proximity could, however, lead to improvements. For women living with abusive husbands, previously acrimonious relations sometimes improved: '… My husband came closer to me and my daughter, taking care of both of us' (30–39 years, married with one child and living with husband in nuclear family; physical, emotional, economic IPV). It was also possible for violence to decrease because of wariness rather than benevolence, when the abuser relied on the respondent: 'My husband fights with me all the time. He constantly bickers. He's not beating me right now because he knows he can't do without me for now' (30–39 years, married with three children and living with husband in joint family; physical, sexual, emotional, economic IPV and family violence).

Nevertheless, most women living with their natal family—away from the source of violence—were positive about it: 'I love taking care of my family and being with them. Glad they support me so much. We're very happy we're getting quality time together' (30–39 years, separated with no children and living with natal family; physical, sexual, emotional, economic IPV).

### Limited mobility
The effects of precarity and confinement were exacerbated by the limits set to mobility. These had two general effects: manifestation of power imbalances and an inability to find relief outside the home. For some women, confinement had '… given my husband more power. It's very difficult for us' (30–39 years, married with four children and living with husband in nuclear family; physical, sexual, emotional, economic IPV). Money was withheld or taken from women and abusers controlled their movements indoors, forced them out of the house, or isolated them socially. 'My husband and in-laws are very abusive. Many times they have locked me up and left me without food' (20–29 years, married with no children and living with husband in joint family; physical, sexual, emotional, economic IPV and family violence). A few survivors said that their partners had not allowed them to access medical care on the pretext of acquiring infection.

Confinement also led to feelings of helplessness and lack of opportunity to decompress. Of the 208 women who had not reported physical violence at first consultation, four said that it had begun during the lockdown: 'My husband beat me for the first time… My husband is getting violent and I'm alone at home and neither of us can leave to cool down. I feel trapped' (20–29 years, married with no children and living with husband in nuclear family; emotional IPV and family violence). Threats of violence from their abusers prevented women from disclosing their unhappiness. This was particularly concerning for respondents in abusive situations in need of counselling services, where the abuser was always around or monitoring their calls. Counsellors sometimes noted reticence in women's answers to questions: 'Client's husband was there and so she was being a little calculating about what

she was saying' (20–29 years, married with three children and living with husband in nuclear family; physical, sexual, emotional, economic IPV). Women were unable to draw on emotional support outside the home from neighbours, friends or family members. 'Before this situation, I spent some time with my neighbours to share my feelings and thoughts, but now due to more housework, I'm not able to manage more time for myself' (30–39 years, married with four children and living with husband in nuclear family; physical, emotional, economic IPV and family violence). Women described hesitancy to disclose their emotions to others for fear of being judged or misunderstood, or because they had limited access to phones. They often felt that it was their responsibility to handle their mental health since the pandemic had already brought enough stress and fear to those around them. 'I don't share with anyone. I don't want to trouble them. I don't talk to my family and I don't have any friends here. So I keep everything to me' (20–29 years, separated with one child and living with natal family; physical, sexual, emotional, economic IPV and family violence).

### Concern for children
Beside their concern for their children's diet, education and socialisation, women were worried that fights, harassment or physical violence that would otherwise have occurred while children were at school or outside the home now happened in front of them. One woman's husband was '…staying at home the whole day … and that has been affecting the mental health of my kids also as they are afraid of the rage that they see after their father is drunk and acts aggressive or hits me' (30–39 years, married with three children and living with husband in nuclear family; physical, sexual, emotional, economic IPV).

### Alcohol
Lockdown changed the availability of alcohol, leading to improvements in some households, but more drinking at home or withdrawal symptoms in others. 'He has been creating a lot of issues being drunk all day and also forces me to have sex sometimes' (30–39 years, living with partner with two children in nuclear family; physical, sexual, emotional, economic IPV). For those reporting improvement, it was primarily through less drinking and more time spent caring for the family. 'My husband has stopped drinking so there is some peace at home' (20–29 years, married with three children in nuclear family; physical, sexual, emotional, economic IPV). In some homes where drinking persisted, not only were tensions heightened, but abusers would leave the house to find alcohol. This caused further worries about exposure to SARS-CoV-2 and added to household fears for health and wellness.

### Women's work
Along with their concerns about basic needs, many women said that their main source of stress was the increase in

household responsibilities that fell disproportionately on them. Some of these related to hygiene: cleaning, disinfection, and childcare in an environment in which viral transmission was a major concern. The extra housework occasioned by more people in the household and children being off school was usually shouldered by women. 'I have extra workload and no help, while if I expect help my husband and mother-in-law abuse me' (30–39 years, married with two children and living with husband in joint family; physical, emotional, economic IPV). Women tended to be tasked with cooking and cleaning and men were tasked with leaving the house to get supplies. The cognitive effects of these responsibilities centred on weariness and frustration with gender norms, when families were '… using the lockdown for harassment and intimidation' (20–29 years, married with no children and living with husband in joint family; physical, sexual emotional, economic IPV and family violence).

Respondents who also worked from home reported difficulty balancing employment with their caring responsibilities, saying that family members had heightened expectations that they would complete chores. 'I have to work more than before and my sister-in-law keeps saying that if I don't work I am just another baggage using their resources' (20–29 years, married with no children and living with husband in joint family; physical, emotional, economic IPV). For these households, domestic labour ranked higher than work and school, and violating expectations could trigger abuse: 'My natal family does emotional violence on me because I work from home, so the full day I am spending time at home, my family expects me to do domestic work' (20–29 years, divorced with one child and in joint family; emotional family violence). Even respondents who had moved in with their natal family said that they felt guilty for burdening them as they were unable to contribute financially, and that their presence added to precarity. 'I am staying with my natal family and now that I am unable to contribute … I feel awkward and a burden to them. Not that they ever verbalised it, but the non-verbal gestures are enough to understand' (40–49 years, married with no children and living in natal family; physical, emotional, economic family violence).

### Addressing needs with survivor-centred approaches

Respondents highlighted several needs during their interviews, many of them being basic survival supplies such as rations and money. Others flagged the need for legal counsel and continued court proceedings to address their domestic violence cases, all of which were interrupted by lockdown. A few needed medical attention and some simply wanted to see their friends and families or take a walk as a moment of solace.

They also discussed their experience with telecounselling. Since all the respondents were survivors of violence with prior connections to counsellors, we asked how telecounselling fit with their circumstances before and during the pandemic. Respondents recognised the importance of distanced counselling services during the pandemic, and generally found telephone discussion a good option. Some actually preferred it to in-person meetings, saying that it overcame geographical and physical barriers and time constraints that had prevented them from visiting a centre. Nevertheless, most said that they would prefer to return to the counselling centre when it was safe. Respondents who reported substantial changes to their daily life said that they were unable to talk at home because of lack of privacy, family members listening in, higher risk of violence if discovered, or limited access to a phone. They said that in-person counselling was better because they felt more comfortable and articulate at the counselling centre and the services would be better. Respondents who did not report increased violence agreed with this, but needed different types of services from the centre, such as psychological counselling for family members other than themselves or legal guidance.

## DISCUSSION

During the response to the COVID-19 pandemic, survivors of domestic violence in Mumbai's urban informal settlements faced—as well as fears of contagion—heightened financial and food insecurity, confinement in small homes in the presence or absence of abusers, coercive control, inability to discuss their experiences with others and an increased burden of domestic labour. Legal processes were to all intents and purposes halted during the pandemic,[8 18] increasing the stress on clients who were waiting for resolution of their cases. Our study explored the needs and priorities of women whose relationships were already marked by intimate partner and family violence, and whose living conditions were already difficult. Despite seeking assistance from our counsellors, they lived in a state of constant fear about finances and food, an increased burden of household responsibilities and tolerance of coercive behaviours.

### Intersectional insecurity

Quarantine measures and restrictions on movement increased financial and food insecurity, linking to pathways of risk via economic constraint and exploitative relationships. All the respondents were already receiving support from our programme. Although unacceptable, domestic violence was an existing feature of their lives on which the effects of lockdown played out. For some, confinement with their partner had led to an increase in physical violence, but their abiding concern was with finances and food security,[22] driven predominantly by loss of employment for either themselves or their family. The matrix of stressors echoes the concerns of intersectional feminism, which encourages broader consideration of social factors that shape survivorship.[40] The increased risk of violence in conditions in which the hardships associated with poverty and informal settlements are augmented is evident in India,[41] rural Bangladesh[42] and in other lower-income countries.[43] We need to pay attention to survivors' basic needs: while some may benefit from exit strategies

and social and legal protective services, there is a clear need for food and financial support.[14] Lack of availability of alcohol (or an increase in drinking at home)[9] led in some cases to increased violence by habitual users. In others, it seems to have diminished irritability and allowed users to exhibit impulse control.[28 29]

## Confinement

Insecurity combines with enforced proximity with perpetrators of violence,[14 20 22 23 44 45] in small homes in urban informal settlements, amplifying the tension between family members and instances of domestic violence.[46] The global discussion tends to equate domestic violence with IPV, but violence by other family members is common in India. We have reported previously that other family members are responsible for around half of emotional violence against women in informal settlements in Mumbai.[32] Lockdown may confine a woman with her husband, but it may equally confine her with her marital family, including members of the extended family who are living together for the duration. This is exacerbated when perpetrators exploit the restrictions associated with the pandemic to exercise power and control,[6 47] including exploiting the fear of contagion.[4 9] Changes in household composition as a result of lockdown altered the situation in which violence occurred. This is an example of how social structure allows (or does not allow) people to translate criminal intentions into action. Routine activity theory suggests that an offence requires an offender, a suitable target and absence of capable guardians, and it is interesting to think about the effects of changes in who is present in the home as changes in a microecology in which domestic violence takes place.[48]

Limited mobility and contact with other people contribute to mental health concerns such as anxiety and depression and further reduce willingness to seek care.[1 44] Lockdown led to decreased freedom and privacy, with attendant physical and psychological stress and fewer sources of social support.[1 9 49] Isolated and unable to seek help,[6 49] women had fewer opportunities to report violence or to escape the situation.[1 4] Although online platforms and mobile applications that enable survivors to chat and seek help have promoted access to support,[50] survivors isolating with family or sharing their devices face constraints to communication by phone,[44] and surveillance of their social media and internet use.[4 47 49] They may also have little time for themselves due to the increase in responsibilities and the small homes common in urban informal settlements. Added to this is some reticence to seek healthcare or alternative shelter for fear of contracting COVID-19.[44] If women do seek care, service provision is itself compromised by closures.[1 9 51]

## Burden of unpaid care

The intersection between women's unpaid care work and violence, underpinned by social norms, has recently been highlighted in India.[52] Gender roles pervade experiences of domestic violence,[7 53 54] predominantly by way of family and social expectations. Sexism organises power in families,[55] the COVID-19 pandemic has increased the need for unpaid care and domestic work,[1 6 17 56] and women's perceived failure to meet the increased family obligations has resulted in abuse.[14]

Feminist political economy suggests that these outcomes were predictable: gendered division of labour in private and public domains define how women can earn a living.[57] The expectation that women perform unpaid domestic labour and that their access to paid work and education is undervalued sets the scene for the increased burden and sources of disagreement over the balance between domestic and remunerated work reported in our study. For many of our clients, the level of housework (often falling entirely on wives and daughters) increased, making it more difficult to complete. Some households divided duties such that women were to remain home while men left for supplies,[58] barring these women from accessing public channels for survivor support amidst an already diminished support landscape.[1 4 9 28] Added to this is the disproportionate effect of the pandemic on opportunities for women and girls, with drastic reductions in women's sources of work in agriculture and the informal sector reported in Nepal and Pakistan.[7 53]

In addition to their immediate health and safety, the needs born of these circumstances also cause concern for the long-term impacts on gender equality, particularly in terms of women's and girls' access to education and independence. This is illustrated by the rapidity with which lockdown returned women and girls to extended family contexts and the prioritising of housework above schoolwork and remunerated work.[59] Pressure to marry and depletion of work opportunities threaten women's educational prospects and their ability to sustain themselves independently.[29 53]

## Limitations

We do not know whether, at population level, the response to the pandemic increased the incidence or intensity of violence against women and girls. All our respondents had suffered physical, sexual, emotional or economic violence and had previously reached a point at which they had consulted a counsellor. They were among the minority of survivors who had made the decision to share their concerns outside the home. This may partly explain the pragmatism that characterised their accounts of the situation during lockdown. They had already begun to think through and enact safety plans and coping strategies and violence was not their primary concern in this new and stressful environment.

Many of the factors influencing survivors' experiences, such as the reasons why some families enduring financial hardship would grow closer and some would experience more tension, are beyond our data. Methodological limitations include the fact that interviews relied on clients' access to phones and sufficient privacy to provide full and accurate details of their situation. Access to both of these is more limited during lockdown and may have

affected the quality of answers given the increased presence of household members. Data quality and depth also varied between interviewers, who were collecting information within their counselling sessions.

## Recommendations

It was and is difficult for policy-makers and civil society organisations to respond to the needs of survivors of violence against women and girls during a pandemic. Our findings highlight the critical role of financial and social support, the need to factor domestic violence into response planning and the importance of support networks.

### Finance and food security are critical

Violence against women intersects with a lack of access to money, food[60] and (for some) shelter. Any response to the current or future threat to public health must prioritise meeting these needs through economic and livelihood support,[42 51 61] simultaneously reducing the need to leave home to find food, rations or work.[25 62] The two key sources of disadvantage are poverty and gender. To address these sources, future strategies must include community leaders in the design and implementation of pandemic management strategies and activities.

### Support networks are important

Our findings underline the central role of support networks that can identify survivors of violence and help them communicate with services.[17] It may be necessary for service providers to pivot to provide individual and group communication electronically rather than face to face. Civil society organisations could expand their remit in two areas. First, preventive counselling to promote healthy relationships and attitudes, beliefs and behaviours among survivors and perpetrators, and foster problem-solving skills. Second, prevention programmes that make communities more aware of their role in identifying and intervening in cases of intimate-partner and domestic violence. It is especially important to take an approach that addresses harmful gendered norms and expectations due to the nature of violence in this context.[52] Current efforts to address domestic violence during the COVID-19 pandemic, including safe-word alert systems, digital or online communications and services, and remote counselling,[50 63] face challenges in our context. For example, a number of respondents were isolated by abusers who used lockdown to control their movement. This prevented them from accessing pharmacies and shops where a safe-word system is typically implemented.

### Domestic violence must be factored into response planning

Government and civil society must prioritise the continued provision of support for survivors of violence against women and girls.[17 42] Women need to know that help is available, how to get it and that their abusers cannot take advantage of social changes to increase the frequency and intensity of violence.[17] This means that services for survivors of violence must be included in national

preparedness, response and recovery plans.[51 56] Responsive funding must be available and be deliverable to the third sector.[51] If programmes are to provide more holistic health protection for survivors of domestic violence, they must involve them in planning. Awareness of the issue is paramount and the media should be enlisted (and step forward, as many have done).[51] Civil society organisations who are engaged in community outreach should continue and augment efforts to mobilise communities around intolerance of violence and include training programmes on violence arising during natural disasters. Examples include training public health and law enforcement officials to understand and identify domestic violence,[64] community mobilisation motivated in part or whole by prevention of violence and allocation of safe shelter for survivors at risk.

## CONCLUSION

In beginning to understand the impact of COVID-19 and subsequent lockdown measures on survivor needs and vulnerabilities, our findings underline the importance of financial precarity and access to resources and support. The response to the epidemic is an increase in existing stressors—particularly poverty and gendered household labour—as well as changes in households and patterns of alcohol use. Programmes must focus on structural vulnerability, distributing resource-based support, prioritising the continued availability of services for survivors of violence and expanding access to support networks. Decision-makers must be aware of the gendered and intersectional effects of interventions and must include survivors of domestic violence in the planning and implementation of public health strategies.

**Acknowledgements** We thank the counsellors who spoke with their clients and collected the information on which the paper was developed. We thank counselling coordinators Jyoti Borkar, Vandana Rokde, Vandana Singh and Shirisha Yeotikar, counselling and legal coordinator Reshma Jagtap, and associate programme director Sangeeta Punekar. Archana Bagra provided financial oversight and Vanessa D'Souza and Shanti Pantvaidya organisational leadership.

**Contributors** ND and TD conceived the study. ND and DO acquired funding. TD and MH developed the methodology. ND, TD, DD and DO oversaw investigation. MH, TD and DO curated the data. MH did the first analysis and wrote the first draft. MH and DO did online supplemental analyses and wrote the revised draft. ND managed the project with support from TD. All authors critically reviewed drafts of the manuscript and read and commented on the final version. DO accepts full responsibility for the work and the conduct of the study, had access to the data and controlled the decision to publish.

**Funding** This work was done by the NIHR Global Health Research Group on a package of care for the mental health of survivors of violence in South Asia, funded by the UK National Institute for Health Research (NIHR 17/63/47).

**Competing interests** ND and TD work with SNEHA, the organisation at which the study was done.

**Patient consent for publication** Not applicable.

**Ethics approval** The project was conducted within the NIHR Global Health Research Group on A package of care for the mental health of survivors of violence in South Asia, with approval for qualitative data collection from the University College London Research Ethics Committee (2744/007) and the Sangath Institutional Review Board (AN_2018_46). Clients gave signed consent for anonymised or pseudonymised use of their data at initial registration. They were

assured of confidentiality, particularly that information would not be shared with their family, community or the media. Counsellors informed clients of their right to access their records, including for use in building legal evidence.

**Provenance and peer review** Not commissioned; externally peer reviewed.

**Data availability statement** Data are available in a public, open access repository. Data are available in Open Science Framework: Osrin, D. (2021, September 7). Covid-19 interviews. Retrieved from osf.io/9kg67

**Open access** This is an open access article distributed in accordance with the Creative Commons Attribution 4.0 Unported (CC BY 4.0) license, which permits others to copy, redistribute, remix, transform and build upon this work for any purpose, provided the original work is properly cited, a link to the licence is given, and indication of whether changes were made. See: https://creativecommons.org/licenses/by/4.0/.

**ORCID iD**
David Osrin http://orcid.org/0000-0001-9691-9684

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
