## [Reviewer comments · BMJ Open]

ARTICLE DETAILS

TITLE (PROVISIONAL)	Intersectional tension: a qualitative study of the effects of the Covid-19 response on survivors of violence against women in urban India
AUTHORS	Huq, Mita; Das, Tanushree; Devakumar, Delan; Daruwalla, Nayreen; Osrin, David

VERSION 1 – REVIEW

REVIEWER	Malta, Monica University of Toronto, Department of Psychiatry
REVIEW RETURNED	18-Mar-2021

GENERAL COMMENTS	The study addresses a highly important public health problem, but this manuscript lacks focus. The authors could base their review on the following critiques/suggestions below. ABSTRACT: please review, it is not clear how your data was collected. Calls to the violence hotline were recorded with consent? Violence survivors were contacted after calling the hotline by the study team? Please explain what 'household composition' means here? The same for the impact of 'household structure' on violence, it is not clear what do you mean. Conclusions are too vague and confusing, please try to address your results and not the overall field of gender-based violence BACKGROUND, 1st paragraph: "...a situation described as a syndemic, danger in danger, a pandemic within a pandemic, a pandemic paradox, or a shadow pandemic." Those concepts need to be explained and not just cited. Those are not synonyms and need to be defined. Overall the introduction lacks focus and is too broad. Try to present the problem you are studying (domestic violence during COVID-19 lockdown) with specific lens on your population, a highly disenfranchised group living bellow the poverty line in settlements. There are other studies conducted with similar populations from Latin America, Africa, Asia - you need to conduct a better and more focused literature review, this will allow you to clearly present your specific research question, population etc. We all know that domestic violence increased during COVID-19 lockdown. What is new about your study? What data/information/singularity/contribution are you bringing to an already crowded field? Focus on this and review your introduction. METHODS: Please describe what was your conceptual framework, if there was any. Also, briefly explain what the framework approach
---

is, its main concepts and strategies before describing what you did within this specific study.

How do you define 'substantive conversations'? Please explain here.

Intercoder reliability was evaluated? How? Please include the results.

RESULTS:

Your study recruited 720 participants, at least a brief SES table should be included (e.g. age, race/ethnicity, number of children, marital status, other available variables)

After each participant quote, consider including a brief SES info instead of the ID code, e.g. single woman aged xx etc. This might help readers put each quote into context.

Try to summarize your results with just a few quotes, the current results section is very confusing and lacks focus. If your key theme is finances/food, stick to this aspect and its possible impact on increased gender-based violence (for instance, the last quote about feeling trapped inside their home doesn't seem to fit here).

The second subtitle should be revised, as you are summarizing women's double burden, and not discussing the impact of 'household composition and responsibilities'. Also, the term 'responsibilities' is highly controversial, why is it assumed by the authors that women should be 'responsible' for household chores? If that's a perception of this social group/study participants, this need to be clarified.

The third subtitle 'domestic violence' also needs to be reviewed. You are including quotes and discussing experiences of domestic violence in all items. Please be aware that domestic violence includes several aspects: physical, sexual, emotional, financial abuse.

Overall all your items are very confusing and overlapping. For instance, you cite financial instability under 'domestic violence', among many other examples. Please read all your interviews, try to organize more suitable themes that actually reflect the experiences, behaviors, views, struggles of this population. A theme as broad as 'domestic violence' is not suitable for a study conducted with domestic violence survivors who are discussing their struggles with domestic violence counsellors. That doesn't make any sense...

Please review all your results section.

DISCUSSION: This section needs a thorough review as well. Try to focus on your results and compare those with existing literature, including new studies published about COVID-19 impact on domestic violence. This is a broad field, and your manuscript will benefit from a better focus and a clear line of thoughts that should guide your introduction, results and discussion.

I'm looking forward to reviewing once again this important study. Thanks for the opportunity to read your study.

REVIEWER	Boxall, Hayley Australian National University
REVIEW RETURNED	14-Apr-2021

GENERAL COMMENTS	Thank you for the opportunity to review this paper. I found it very interesting, and believe with some additional work will make a valuable contribution to the space. I have made specific comments below, but in general my main queries/recommendations are this:  • The study situates itself as addressing a key gap in knowledge, primarily that very little is known about the impact of COVID on domestic violence, and the factors associated with COVID-19 that may contribute to changes in domestic violence. However, this is simply not true anymore. Since data collection was finalised for this study, there has been a huge influx of papers on this topic using various methodologies. Therefore, situating this paper as addressing this knowledge gap is inaccurate – it contributes to a growing body of evidence about COVID-19 and domestic violence, a unique dimension being its direct engagement with victims/survivors, focus on understanding service needs and broader focus on DV rather than IPV. • The methodology and definitions for this study are currently vague at best, and I have raised specific concerns below about the reference to ‘duplicates’ in the sample, and that women were being observed by their partner at time of completing the interview. • Conflict of interest – the authors declared no conflict of interest in undertaking this research. However, it becomes apparent that at least one if not all of the authors are associated with the Program through which recruitment for the study was undertaken. The final conclusion for this study, the need for additional and longer-term services integrated with response planning, I feel is a fair one and certainly supported by not only this study but multiple others. However, I believe that the authors should declare a conflict of interest in their ongoing engagement with a front-line service that would be a direct beneficiary of any policies that were enacted as a result of the findings from this study. Abstract:  • Objectives: ‘affected their vulnerabilities’ – unclear what this means, exacerbated or impacted pre-existing factors that increased their likelihood of experiencing IPV • Background: Considering the timing of this study, it is appropriate to reference the now extensive literature which has identified consistent high rates of DV using self-report data, including increases in frequency and intensity of violence during the initial stages of the COVID-19 pandemic see https://www.cgdev.org/publication/violence-against-women-and-children-during-covid-19-one-year-and-100-papers-fourth; • ‘An increase in violence against women was, however, well-recognised in the media and a small number of studies, but the breadth of literature exploring how and why survivors experience domestic violence during the Covid-19 pandemic has so far been limited.²⁰ This may have been true initially but not anymore, and the reference itself is dated (June 2020). See above for suggested resources. • There is general over-reliance on the term ‘vulnerabilities’ with no clarification for what this means. Setting:  • It becomes apparent in the second last sentence of the first para with the change to possessive term ‘we’ that at least one of the authors works at the Program for Prevention of VAW. This should be noted up-front as a potential conflict of interest
---

	Data collection  • ‘Counsellors provided clients with information about COVID-19?’ What information? • In subsequent sections of the paper (eg p8) it is noted that some respondents were reticent about providing some information because their partner was there with them. This is not noted in any place that I can see as a major limitation of this study – that women may be observed and so may not provide full answers. Also, what were the safety protocols embeded to protect the safety of women whose partners were in the room? I find it really concerning that they continued to converse with participants and ask them questions for the research knowing they were being observed. Data analysis  • The authors used framework analysis but no justification is given for why this approach was selected. • I think this section of the paper would benefit from including a description of framework analysis to orientate unfamiliar readers. • The reference to duplicates is concerning, the implication being that some clients were interviewed more than once. Considering the significant risks associated with contacting women during lockdown periods I find it concerning that there was no protocol in place to minimise the likelihood of duplicates entering the sample – how did this happen? Results  • What was the sample size after removal of duplicates? Household size and composition  • This sentence is a bit unclear – ‘The effects of changes in household structure and function developed on the background of insecurity in terms of basic needs’. Suggest rewording for clarity • ‘...but three general situations’ – suggest rewording, ‘three household structures were commonly described’ • ‘and the experience was not new’. What experience? • The cognitive effects of these responsibilities centred on weariness and frustration with gender norms, when families were “... using the lockdown for harassment and intimidation” (ID 29, 19 April 2020). This is a bit unclear but seems to suggest that family members were capitalising on the pandemic conditions to further abuse women? However I don’t think you have really stepped out this link (if this is what you are suggesting). Would suggest expanding. • ‘Reproductive labour ranked higher than productive labour and expectations could be triggers for abuse’. Not sure what is meant here – is the suggestion that women were experiencing more pressure to have children? From the follow-up quote it seems that the point is that women’s involvement in domestic unpaid labour was expected and when women failed to live up to familial expectations around this because of paid employment, they would experience abuse. Need to clarify. Domestic violence  • 22% experienced domestic violence. What is your definition of domestic violence? There is no provided definition and not sure what is included here, although you reference behaviours consistent with family violence and child abuse, as well as emotional abuse, economic abuse and physical violence specifically. • Would be worth clarifying what the ‘observation period’ was for this statistic. If interviews conducted April-July and lockdown occurred in March, follow-up ranged from 2 weeks to 4.5 months? • As a broader point, you have been commenting on respondent’s experiences of domestic violence throughout the results section
--	--

	and only now have used the phrase domestic violence and reported the statistic. It's a bit jarring and seems to come out of nowhere. I would suggest including this statistic in the Results first para, including your definition of domestic violence.  • This section is primarily about the impact of the lockdown conditions on patterns of pre-existing domestic violence – both positive and negative. I would change the title of this section to 'Changes in patterns of domestic violence' to make it clearer. • 'It was possible for violence to decrease as a result of wariness rather than benevolence when the abuser relied on the respondent'. This is a VERY interesting finding which I think is brushed over slightly. Is it wariness or is it that the costs associated with the offending may have increased due to changes in the dynamics within the relationship that are attributable to COVID-19? Routine activities theory and rational choice theory could provide a useful lens for understanding this finding. • The authors attribute a change in DV patterns to a couple of factors, including 'exposure to the abuser(s), higher stress levels ("tension"), and heightened expectations'. The first one is particularly interesting considering new research which has shown that time spent at home in and of itself is not associated with DV. • 'Although all the respondents had experienced domestic violence, some experienced 207 physical abuse for the first time' This is another finding which is brushed over very briefly but is actually really important. Can you expand on this? Discussion  • 'gendered changes in household responsibilities and pressures, 273 and changes in the forms of domestic violence' Were they changes in responsibilities or increased responsibilities? • 'A third concern, and one that we have expressed before, is that the global discussion tends to equate domestic violence with intimate partner violence. In our setting, although most physical and sexual violence is perpetrated by partners, around half of emotional violence is done by other family members'. This is a really important and fantastic point. However I don't think the authors have done enough in the analysis to draw out the different forms of violence occurring within households. It would be useful to include a section that draws out the family violence women reported, as distinct from IPV. • 'The response to the epidemic is an increase in existing stressors' – Do the authors mean 'The response to the pandemic has led to an increase in existing stressors'?
--	--

VERSION 1 – AUTHOR RESPONSE

Reviewer 1

The study addresses a highly important public health problem, but this manuscript lacks focus. The authors could base their review on the following critiques/suggestions below.

We have restructured the whole paper, including the title, in order to provide more focus and clarity.

ABSTRACT: please review, it is not clear how your data was collected. Calls to the violence hotline were recorded with consent? Violence survivors were contacted after calling the hotline by the study team?

We have done this:

“During telephone follow-up with registered survivors of violence against women with whom they already had a relationship, counsellors asked about...”

Please explain what ‘household composition’ means here? The same for the impact of ‘household structure’ on violence, it is not clear what do you mean.

We have removed these terms in the revised abstract.

Conclusions are too vague and confusing, please try to address your results and not the overall field of gender-based violence.

In keeping with our revision to give the manuscript more focus, the conclusion now reads:

“The Covid-19 pandemic has increased the burden of poverty and gendered unpaid care. Finance and food security are critical considerations for future response, which should consider intersectional inequality, financial support, prioritising the continued availability of services for survivors of violence, and expanding access to social networks. Decision-makers must be aware of the gendered and intersectional effects of interventions and must include residents of informal settlements who are survivors of domestic violence in the planning and implementation of public health strategies.”

BACKGROUND, 1st paragraph: “...a situation described as a syndemic, danger in danger, a pandemic within a pandemic, a pandemic paradox, or a shadow pandemic.” Those concepts need to be explained and not just cited. Those are not synonyms and need to be defined.

We have rewritten the background and removed these terms.

Overall the introduction lacks focus and is too broad. Try to present the problem you are studying (domestic violence during COVID-19 lockdown) with specific lens on your population, a highly disenfranchised group living below the poverty line in settlements. There are other studies conducted with similar populations from Latin America, Africa, Asia - you need to conduct a better and more focused literature review, this will allow you to clearly present your specific research question, population etc. We all know that domestic violence increased during COVID-19 lockdown. What is new about your study? What data/information/singularity/contribution are you bringing to an already crowded field? Focus on this and review your introduction.

We have rewritten the introduction with these suggestions in mind.

METHODS: Please describe what was your conceptual framework, if there was any.

We have added a discussion of the conceptual frameworks that informed our analysis: intersectional feminism and feminist political economy. We first did this in the Methods section, as suggested by the reviewer. We realised, however, that it seemed better to align the findings with conceptual frameworks in the Discussion section because readers could intuitively see the connection.

Also, briefly explain what the framework approach is, its main concepts and strategies before describing what you did within this specific study.

We have done this:

“We used the framework approach to thematic analysis. Increasingly used in health research, this approach lends itself to studies by multidisciplinary teams with fairly specific objectives and questions, limited time for data collection, and a predefined large sample. Data are charted in a matrix in which

rows represent interviewees and columns represent codes, and information can be compared across cases and codes. We identified some of these codes before the analysis and augmented them with codes that emerged.”

How do you define ‘substantive conversations’? Please explain here.

We have removed this term.

Intercoder reliability was evaluated? How? Please include the results.

“The two authors who analysed the data compared their codes iteratively. The emergent themes were discussed over four iterations, with inputs from all authors.”

RESULTS:

Your study recruited 720 participants, at least a brief SES table should be included (e.g. age, race/ethnicity, number of children, marital status, other available variables)

We revisited the dataset in light of this comment. The revised version used for the analysis draws on a dataset of 586 respondents for whom we had full information on their initial consultation before the study so that we could contextualise their experience. We have added a summary table and information on the background to consultation in terms of the forms of violence they were experiencing and their home situations.

After each participant quote, consider including a brief SES info instead of the ID code, e.g. single woman aged xx etc. This might help readers put each quote into context.

We liked this suggestion very much and have added information after each quote. Actually, these additions are quite long, but we think that they enrich the reader’s understanding of the situation.

Try to summarize your results with just a few quotes, the current results section is very confusing and lacks focus. If your key theme is finances/food, stick to this aspect and its possible impact on increased gender-based violence (for instance, the last quote about feeling trapped inside their home doesn’t seem to fit here).

Thanks for this comment. We have restructured and edited the Results section substantially.

The second subtitle should be revised, as you are summarizing women’s double burden, and not discussing the impact of ‘household composition and responsibilities’. Also, the term ‘responsibilities’ is highly controversial, why is it assumed by the authors that women should be ‘responsible’ for household chores? If that’s a perception of this social group/study participants, this need to be clarified.

Again, thanks: we have reordered and restructured the Results section and changed the subtitles.

The third subtitle ‘domestic violence’ also needs to be reviewed. You are including quotes and discussing experiences of domestic violence in all items. Please be aware that domestic violence includes several aspects: physical, sexual, emotional, financial abuse.

We have reordered and restructured the Results section and changed the subtitles, as well as adding information after each quote.

Overall all your items are very confusing and overlapping. For instance, you cite financial instability under ‘domestic violence’, among many other examples. Please read all your interviews, try to organize more suitable themes that actually reflect the experiences, behaviors, views, struggles of this population. A theme as broad as ‘domestic violence’ is not suitable for a study conducted with

domestic violence survivors who are discussing their struggles with domestic violence counsellors. That doesn't make any sense...

Thanks. We have reordered and restructured the Results section, making sure that the narrative is clear and organising themes into an intuitive sequence.

DISCUSSION: This section needs a thorough review as well. Try to focus on your results and compare those with existing literature, including new studies published about COVID-19 impact on domestic violence. This is a broad field, and your manuscript will benefit from a better focus and a clear line of thoughts that should guide your introduction, results and discussion.

We have restructured the Discussion section, reflected in a more organised way against the existing literature and making reference to theory. We have updated our review to include studies published since the paper was first written.

Reviewer 2

- The study situates itself as addressing a key gap in knowledge, primarily that very little is known about the impact of COVID on domestic violence, and the factors associated with COVID-19 that may contribute to changes in domestic violence. However, this is simply not true anymore. Since data collection was finalised for this study, there has been a huge influx of papers on this topic using various methodologies. Therefore, situating this paper as addressing this knowledge gap is inaccurate – it contributes to a growing body of evidence about COVID-19 and domestic violence, a unique dimension being its direct engagement with victims/survivors, focus on understanding service needs and broader focus on DV rather than IPV.

Thanks. We have substantially rewritten all the sections of the paper, including what we hope is an up-to-date reading of publications. We have emphasised the three points in a way that we hope readers now find clear.

- The methodology and definitions for this study are currently vague at best, and I have raised specific concerns below about the reference to 'duplicates' in the sample, and that women were being observed by their partner at time of completing the interview.

We have provided more information in the Methods section. We understand that the reference to duplicates was confusing. Each client was interviewed only once. What we meant was that some lines of the dataset had been duplicated accidentally by counsellors at data entry and we had cleaned it to fix this. We have removed the statement because, actually, removing duplicate entries does not amount to a filtering of the data. However... the reviewers' comments in general stimulated us to make the whole thing cleaner by matching the interview records with previous information about clients at the time of registration. We achieved this comprehensively in 586 cases and it allowed us to provide summaries of the duration of counselling, home situation, and forms of violence women were experiencing. We hope that the reviewer agrees that this has improved the paper.

- Conflict of interest – the authors declared no conflict of interest in undertaking this research. However, it becomes apparent that at least one if not all of the authors are associated with the Program through which recruitment for the study was undertaken. The final conclusion for this study, the need for additional and longer-term services integrated with response planning, I feel is a fair one and certainly supported by not only this study but multiple others. However, I believe that the authors should declare a conflict of interest in their ongoing engagement with a front-line service that would be a direct beneficiary of any policies that were enacted as a result of the findings from this study.

We appreciate this comment. Daruwalla and Das work at the organisation at which the study was done. We don't believe this constitutes a conflict of interest according to the BMJ guidelines, but we have entered a statement to this effect and are happy to follow the editor's advice.

Abstract:

- Objectives: 'affected their vulnerabilities' – unclear what this means, exacerbated or impacted pre-existing factors that increased their likelihood of experiencing IPV

We have removed this phrase.

- Background: Considering the timing of this study, it is appropriate to reference the now extensive literature which has identified consistent high rates of DV using self-report data, including increases in frequency and intensity of violence during the initial stages of the COVID-19 pandemic see <https://eur01.safelinks.protection.outlook.com/?url=https%3A%2F%2Fwww.cgdev.org%2Fpublication%2Fviolence-against-women-and-children-during-covid-19-one-year-and-100-papers-fourth&data=04%7C01%7C%7Ccaf0d12eda33418bcf9a08d9034557ea%7C1faf88fea9984c5b93c9210a11d9a5c2%7C0%7C0%7C637544418627826314%7CUnknown%7CTWFpbGZsb3d8eyJWIjojMC4wLjAwMDAiLCJQIjoiV2luMzliLCJBTiI6Iik1haWwiLCJXVCi6Mn0%3D%7C1000&data=y%2BcjZZIWV7oLVgne1YvHfg%2B9r77Hq47rKDDjD%2B1nhwA%3D&reserved=0;>

- 'An increase in violence against women was, however, well-recognised in the media and a small number of studies, but the breadth of literature exploring how and why survivors experience domestic violence during the Covid-19 pandemic has so far been limited.²⁰ This may have been true initially but not anymore, and the reference itself is dated (June 2020). See above for suggested resources.

Thanks for the reference. We have updated the literature review and included new information in the paper.

- There is general over-reliance on the term 'vulnerabilities' with no clarification for what this means.

We agree and have removed the word wherever it was used.

Setting:

- It becomes apparent in the second last sentence of the first para with the change to possessive term 'we' that at least one of the authors works at the Program for Prevention of VAW. This should be noted up-front as a potential conflict of interest

We hope we have addressed this above.

Data collection

- 'Counsellors provided clients with information about COVID-19?' What information?

We have added the following:

"Counsellors provided all clients with information on Covid-19 (transmission, social distancing, handwashing, masks, emotional hygiene, anxiety management, freedom to contact the organisation for advice)."

- In subsequent sections of the paper (eg p8) it is noted that some respondents were reticent about providing some information because their partner was there with them. This is not noted in any place that I can see as a major limitation of this study – that women may be observed and so may not provide full answers. Also, what were the safety protocols embedded to protect the safety of women whose partners were in the room? I find it really concerning that they continued to converse with participants and ask them questions for the research knowing they were being observed.

We think that the previous version of the paper must have not made the study context clear enough. All the women involved were survivors of violence who were already being seen by a counsellor. The counsellors followed up to offer telephone contact for consultations during the lockdown. During a consultation, they asked questions that informed the study. These questions were nested within a protocolised counselling session. The counsellors were not researchers: they were trained specialists in violence against women. We hope that our rewriting of the Methods section makes this clear.

Data analysis

- The authors used framework analysis but no justification is given for why this approach was selected.
- I think this section of the paper would benefit from including a description of framework analysis to orientate unfamiliar readers.

We have added the following:

“We used the framework approach to thematic analysis. Increasingly used in health research, this approach lends itself to studies by multidisciplinary teams with fairly specific objectives and questions, limited time for data collection, and a predefined large sample. Data are charted in a matrix in which rows represent interviewees and columns represent codes, and information can be compared across cases and codes. We identified some of these codes before the analysis and augmented them with codes that emerged. An indexing and data exploration process in NVivo 12.06.0 expanded this framework by inductively adding themes and sub-themes.”

- The reference to duplicates is concerning, the implication being that some clients were interviewed more than once. Considering the significant risks associated with contacting women during lockdown periods I find it concerning that there was no protocol in place to minimise the likelihood of duplicates entering the sample – how did this happen?

We have provided more information in the Methods section. We understand that the reference to duplicates was confusing. Each client was interviewed only once. What we meant was that some lines of the dataset had been duplicated accidentally at data entry and we had cleaned it to fix this. We have removed the statement because, actually, removing duplicate entries does not amount to a filtering of the data. However... the reviewers' comments in general stimulated us to make the whole thing cleaner by matching the interview records with previous information about clients at the time of registration. We achieved this comprehensively in 586 cases and it allowed us to provide summaries of the duration of counselling, home situation, and forms of violence women were experiencing. We hope that the reviewer agrees that this has improved the paper.

Results

- What was the sample size after removal of duplicates?

720, although see the comment above which describes the revised sample of 586.

Household size and composition

- This sentence is a bit unclear – ‘The effects of changes in household structure and function developed on the background of insecurity in terms of basic needs’. Suggest rewording for clarity

We have removed this in the rewriting.

- ‘...but three general situations’ – suggest rewording, ‘three household structures were commonly described’

We have removed this in the rewriting and provided more information on where women were living.

- 'and the experience was not new'. What experience?

We have removed this.

- The cognitive effects of these responsibilities centred on weariness and frustration with gender norms, when families were "... using the lockdown for harassment and intimidation" (ID 29, 19 April 2020). This is a bit unclear but seems to suggest that family members were capitalising on the pandemic conditions to further abuse women? However I don't think you have really stepped out this link (if this is what you are suggesting). Would suggest expanding.

Thanks. That is what we meant and we have expanded on it:

"The effects of precarity and confinement were exacerbated by the limits set to mobility. These had two general effects: manifestation of power imbalances and an inability to find relief outside the home. For some women, confinement had "... given my husband more power. It's very difficult for us" (30-39 years, married with 4 children and living with husband in nuclear family; physical, sexual, emotional, economic IPV). Money was withheld or taken from women and abusers controlled their movements indoors, forced them out of the house, or isolated them socially. "My husband and in-laws are very abusive. Many times they have locked me up and left me without food" (20-29 years, married with no children and living with husband in joint family; physical, sexual, emotional, economic IPV and family violence). A few survivors said that their partners had not allowed them to access medical care on the pretext of acquiring infection."

- 'Reproductive labour ranked higher than productive labour and expectations could be triggers for abuse'. Not sure what is meant here – is the suggestion that women were experiencing more pressure to have children? From the follow-up quote it seems that the point is that women's involvement in domestic unpaid labour was expected and when women failed to live up to familial expectations around this because of paid employment, they would experience abuse. Need to clarify.

Yes, the latter is what we meant. The comment made us realise that some readers will not be familiar between the distinction between productive and reproductive labour

(https://en.wikipedia.org/wiki/Reproductive_labor). We have removed the phrase and replaced it with terms such as 'unpaid care work'.

Domestic violence

- 22% experienced domestic violence. What is your definition of domestic violence? There is no provided definition and not sure what is included here, although you reference behaviours consistent with family violence and child abuse, as well as emotional abuse, economic abuse and physical violence specifically.

Thanks. We hope that the first section of the Results now makes this clear. We see domestic violence as violence by an intimate partner or other family member that may be physical, sexual, emotional, economic, coercive control, or neglect (for contextual reasons, we don't tend to include FGM and trafficking). We usually include formal definitions of violence against women and domestic violence in our publications, but in this case we didn't think it was necessary. We can add them...

- Would be worth clarifying what the 'observation period' was for this statistic. If interviews conducted April-July and lockdown occurred in March, follow-up ranged from 2 weeks to 4.5 months?

We have added to the Results section:

"At the time of interview, 54% of respondents were in the first month of lockdown, 40% in the second, and 6% in the third month or later."

- As a broader point, you have been commenting on respondent's experiences of domestic violence throughout the results section and only now have used the phrase domestic violence and reported the statistic. It's a bit jarring and seems to come out of nowhere. I would suggest including this statistic in the Results first para, including your definition of domestic violence.

Thanks. We have done this in the new first paragraph of Results.

- This section is primarily about the impact of the lockdown conditions on patterns of pre-existing domestic violence – both positive and negative. I would change the title of this section to 'Changes in patterns of domestic violence' to make it clearer.

We have restructured the Results section with different subheadings.

- 'It was possible for violence to decrease as a result of wariness rather than benevolence when the abuser relied on the respondent'. This is a VERY interesting finding which I think is brushed over slightly. Is it wariness or is it that the costs associated with the offending may have increased due to changes in the dynamics within the relationship that are attributable to COVID-19? Routine activities theory and rational choice theory could provide a useful lens for understanding this finding.
- The authors attribute a change in DV patterns to a couple of factors, including 'exposure to the abuser(s), higher stress levels ("tension"), and heightened expectations'. The first one is particularly interesting considering new research which has shown that time spent at home in and of itself is not associated with DV.

These are really interesting points. We teach routine activities theory in our seminars on urban health, but had not seen it applied to domestic violence. We have added a comment in the Discussion section. The paper has already got longer and we hope to think about the ideas more in subsequent work.

"Changes in household composition as a result of lockdown altered the situation in which violence occurred. This is an example of how social structure allows (or does not allow) people to translate criminal intentions into action. Routine activity theory suggests that an offence requires an offender, a suitable target, and absence of capable guardians, and it is interesting to think about the effects of changes in who is present in the home as changes in a micro-ecology in which domestic violence takes place."

- 'Although all the respondents had experienced domestic violence, some experienced physical abuse for the first time' This is another finding which is brushed over very briefly but is actually really important. Can you expand on this?

We have analysed this more. Of 208 women who did not report physical violence at first consultation, 20 (10%) reported some form of domestic violence during lockdown. This was physical violence in 4 cases. We have added to the Results section:

"Confinement also led to feelings of helplessness and lack of opportunity to decompress. Of the 208 women who had not reported physical violence at first consultation, four said that it had begun during the lockdown: "My husband beat me for the first time... My husband is getting violent and I'm alone at home and neither of us can leave to cool down. I feel trapped" (20-29 years, married with no children and living with husband in nuclear family; emotional IPV and family violence)."

Discussion

- 'gendered changes in household responsibilities and pressures, and changes in the forms of domestic violence' Were they changes in responsibilities or increased responsibilities?

We meant increased responsibilities. We appreciate that this was vague and hope that the rewritten section is now clear.

- ‘A third concern, and one that we have expressed before, is that the global discussion tends to equate domestic violence with intimate partner violence. In our setting, although most physical and sexual violence is perpetrated by partners, around half of emotional violence is done by other family members’. This is a really important and fantastic point. However I don’t think the authors have done enough in the analysis to draw out the different forms of violence occurring within households. It would be useful to include a section that draws out the family violence women reported, as distinct from IPV.

We have added to the Results section:

“Family violence was predominantly emotional (87%), economic (63%), and physical violence (54%), and coercive control (46%).”

“Women living in joint families faced predominantly emotional violence, and physical violence in three cases.”

- ‘The response to the epidemic is an increase in existing stressors’ – Do the authors mean ‘The response to the pandemic has led to an increase in existing stressors’?

Yes, although we have rewritten this section.

VERSION 2 – REVIEW

REVIEWER	Boxall, Hayley Australian National University
REVIEW RETURNED	12-Jun-2021
GENERAL COMMENTS	Thank you for providing me with an opportunity to review this paper again. The changes made by the author have improved this paper significantly - I appreciate the careful attention that has been paid to my own and the other reviewer's comments. The paper makes a valuable contribution to current knowledge about the impact of COVID-19 on domestic and family violence, and is a very interesting read as well.